# Online-Network-Group Use and Political-Participation Intention in China: The Analysis Based on CSS 2019 Survey Data

**DOI:** 10.3390/bs13040302

**Published:** 2023-04-03

**Authors:** Zipeng Li, Fengrong Liu

**Affiliations:** The School of Public Policy and Management, Tsinghua University, Beijing 100084, China

**Keywords:** Internet use, online-network group, political-participation intention, effect study

## Abstract

Although there are many studies discussing the effect of Internet use on political participation, the literature has rarely focused on the relationship between the use of online-network groups and the political-participation intention in contemporary China. The discussion of this relationship is significant, as it offers a new perspective on reviewing the mobilization theory of media, especially in the context of online-network groups, and potentially provides a new channel for mobilizing a wider range of people for politics once the relationship is significant. This study aims to answer the following question: Can we predict the political-participation intention of Chinese citizens through the use of online-network groups? Based on the data of the China Social Survey 2019, this study uses the hierarchical logistic-regression method. The research finds that the specific online-network groups that can predict political-participation intention are mainly concentrated in the category of emotional relationships. Among them, although most of the online-network groups are positively correlated with political-participation intention, the possibility of generating political-participation intention of those who join the relative network group is significantly lower than for those who do not. The virtual connection built by online communication technology, the social relations, and the influence of social groups on individuals can help to explain the correlation between them.

## 1. Introduction

The rapid development of information technology brings a new social form guided by the logic of networks [1]. The Internet not only penetrates almost all aspects of contemporary Chinese life but also affects the model of people’s thinking. In June 2022, the number of Internet users in China reached 1.051 billion, with the Internet penetration rate 74.4% [2]. Both the scale of Internet users and the Internet-penetration rate have achieved sustained growth in China for 10 consecutive years. This rapid development has attracted scholars’ continuous attention to the impacts of the development of the Internet on Chinese society. Among various impacts, the impact of Internet use on citizens’ political participation has triggered heated discussions.

As a continuation of the classic media-effect research in the Internet era, the effects of Internet use on political participation in the U.S. and other Western countries have long attracted the attention of scholars. Verba and Nie defined political participation as “those activities by private citizens that are more or less directly aimed at influencing the selection of governmental personnel and/or the actions they take” with four modes of organized participation: voting, campaign activity, cooperative activity, and citizen-initiated contacts [3]. According to Huntington’s understanding, the term “political participation” refers to an activity that affects the government’s decision-making by actors spontaneously or mobilized by others [4]. The changing political contexts and behaviors continuously expand the boundary of this concept, which incorporates broader connotation. At the start of the 21st century, van Deth summarized four features of political participation, including (1) It contains *activity*, (2) it is *voluntary*, (3) it refers to people as *non-professionals* or *amateurs*, and (4) it concerns *government, politics, or the state* [5]. With the continuous expansion of political participation, a new taxonomy is introduced to cover different forms of participation that is “creative, expressive, individualized and digitally enabled” [6]. Both old and new forms of participation are incorporated into this taxonomy, covering “voting, digital networked participation, institutionalized participation, protest, civic participation and consumerist participation” [6]. 

There is still debate on how the use of the Internet affects citizens’ political participation. According to the mobilization theory, information technology reduces the cost of communication and improves democratic citizenship. Scholars who support this theory believe that there is a positive correlation between network use and civil political participation [7,8,9]. Research using the data of the national elections of the U.S. argues that the use of the Internet promotes citizens to participate more actively in political activities [10]. Some scholars have confirmed this correlation in the context of social media such as MySpace and Facebook [11]. However, the limitation of the mobilization theory is that it originated in the context of Western capitalist democratic countries, which may set constraints for its universal applicability. Some other studies focusing on this relationship have denied the mobilization theory. A series of studies has found that there is a very limited correlation between Internet use and political participation [10,12]. As a representative not supporting the mobilization theory, Robert Putnam believes that as the Internet is a technology mainly used for entertainment, it has a harmful impact on political participation [13]. His subsequent research found that Americans who read online news were less likely to participate in politics than others [14]. Delli-Carpini also has a similar view: that the increase in Internet use causes a reduction in the time spent on political participation [15]. Some other scholars have found that the use of the Internet would lead to the dispersion of users’ energy, which leaves the public little time to participate in politics [16]. 

Recently published studies have further investigated the complexity and subtlety of the political implications of the Internet. By analyzing the survey results of a German population, scholars developed a multi-dimensional taxonomy of political participation in which “digitally networked participation” is an important type that cannot be ignored [5]. Based on cross-national data from the United States, the United Kingdom, and France, researchers found that political participation enacted by social media shows distinctions from traditional participation in the factors of structure [17]. The political effect of the Internet is complex, as it is inconsistent in terms of the various types and different time periods of political participation [18]. With the popularity of social media, there are also studies focusing on revealing the specific effects of social-media exposure. By analyzing data from a two-wave panel study, one study found that political social-media exposure showed a distraction effect, as it was positively correlated with low-effect political participation [19]. 

## 2. Literature Review 

The debate about the political impact of the Internet continues in the context of contemporary China. Zhongdang Pan found that the use of the Internet has a nationwide positive impact on the political participation of Chinese citizens, but there are regional and local differences related to the level of economic development, the degree of urbanization, and the level of residents’ education [20]. Leizhen Zang found that the use of new media and the openness of the political-opportunity structure are significantly correlated to political participation, but their impacts on political participation are relatively limited [21]. With the popularity of social media in China, more and more studies have focused on its political impact. Some scholars believe that the low cost and convenience of social media for communication has greatly enhanced the initiative and enthusiasm of Chinese youth in political participation [22]. It is also argued that social media not only promotes the online political participation of Chinese youth but also strengthens their offline political participation [23]. From a comparative perspective of mainland China, Hong Kong, and Taiwan, one study found that sharing political information and connecting with public actors could continuously predict the offline and online political participation of social-media users [24]. Although most studies held a positive view, some scholars had opposite opinions. Lidan Chen and others believe that the popularity of social media reduces the level of people’s political participation [25]. 

It is worthwhile to note that the people who engage in political behavior cannot reach all citizens. Due to institutional and practical constraints, there are problems with electoral political participation in China [26]. Ordinary Chinese citizens have only limited opportunities to participate in formal politics [27]. Among those who do not participate in politics, some may have the intention of political participation. Scholars have found that both media reports and a supportive community can predict people’s intention of political participation. Based on the survey conducted prior the 2012 South Korean presidential election, research verified the direct association between the media’s report of the nominee Jae In Moon being behind in the polls and the political-participation intention of his supporters [28]. By surveying young adults living in the former East Germany in 2010, one study found that a supportive community could predict political-participation intention. However, neither a supportive family nor supportive friends and acquaintances were significantly associated with the intention of political participation [29]. Although there are some findings of factors that can predict political-participation intention, very few studies have strictly defined this concept. Referring to the definition of “political participation” [4] and the available data from the China Social Survey 2019, this paper defines “political-participation intention” as whether those who do not participate in the activities that affect government decision-making are willing to participate in the activities that influence political decisions. The study of political-participation intention may not be less important than the study of actual political behavior. Everyone can decide to what extent they are willing to participate in activities that affect government decision-making. The intention of participating in politics can provide a new perspective of understanding the possible political effects of using the Internet in contemporary China. Among the few studies focusing on the intention to participate in politics, scholars have found that reading online political news had a significant negative correlation with the willingness to participate in the election in Taiwan [30]. However, the geographic focus means that its conclusions cannot be generalized to mainland China. Our understanding of the relationship between Internet use and political-participation intention in mainland China is still very limited.

From the perspective of online media, existing research has had a narrow focus on the use of the Internet, mainly referring to the dimensions of reading online news [20,30], using online social media [22,23,24,25], and so on. In addition to these means of using the Internet, there are many other popular uses of the Internet in Chinese daily life. Network groups (*wangluo quanqun*) are one of the most popular ways of using the Internet in China, but few studies have explored their political effects. The meaning of this term is a group or circle of people established through the Internet. In this group, people can communicate with each other through the Internet, and others outside of the group cannot access the information posted inside the group. Network groups include WeChat, QQ, and Weibo, which include other popular Internet functions in China. The utilization rate of instant-messaging applications such as WeChat and QQ in China reached 90.7% in 2020, with 80.5% of WeChat users joining various groups of friends [31]. About 88.7% of the respondents used WeChat group and QQ groups to discuss and obtain current political news, current-affairs comments, and other information [32]. The significance of focusing on network groups for communication research is that the network group has built a new connection mode between network nodes, which highlights the changes in the connection mode between people with the development of mobile Internet technology. Lan Peng argued that network groups are a manifestation of “network communities,” which have characteristics of closeness and homogeneity [33]. With the organizational relationship being flat, group members have relatively equal opportunities to speak. Network groups have the characteristics of a social network, as they can maintain both strong and weak relationships. Individuals can switch quickly and frequently between different network groups [34]. Studying online-media use from the perspective of network groups offers an opportunity to study the relationship between Internet use and the willingness of political participation from the perspective of the social interaction of specific groups in an online network.

Although few studies have explored network groups from the perspective of political effects, some scholars have paid attention to network-group research from the perspective of cultural characteristics, interest expression, and social-network characteristics. From the perspective of the cultural characteristics of a network group, Wenbo Kuang understood a network group as a “virtual aggregation space” representing new social relations with the cultural characteristics of organization, isolative exclusiveness, and antagonism [35]. There are also studies from the perspective of interest expression. It is argued that a “group” on social media combines interaction and communication between teenagers to form an interest-expression space with more freedom [23]. Joining a network group is a popular way for Chinese people to socialize. Research has found that by joining various network groups, network communication presents the characteristics of “circlization (*quan ceng hua*)”: It is easier for information to flow horizontally in the same layer psychologically, whereas the closeness between groups blocks the information flow across layers [33]. Among studies of network groups, there is still a lack of research on the dimension of political effects of the use of network groups.

Apart from cultural characteristics, interest expression, and social-network characteristics, another important perspective for approaching online-network groups is the emotion curated between each user and other members inside the group. The literature reveals that emotion and politics attract scholars’ interest, especially the relationship between emotion and democratic views [36], information process [37], political polarization [38], and so on [39]. On the politicians’ side, emotion can be an effective tool for their political purposes and goals. By using critical-discourse analysis to analyze the speeches of politicians, researchers argue that emotional and rival narratives are used by politicians to polarize political opinions [40]. Based on the corpora of press releases of the United Kingdom Independence Party (UKIP) and Labour Party and the qualitative evidence of how specific emotions are triggered by these two parties, research has revealed different “affective-discourse practices” in which the UKIP invokes more positive emotions while legitimizing fear and anger and Labour addresses social worry and concerns characteristically by framing reactions [41]. Citizens’ perception of emotion has also attracted scholars’ attention. By analyzing evidence from the Mass Observation Archive, scholars have found that although reason and emotion cannot be separated in political decisions, citizens perceive them as competing with each other [42]. As many online-network groups are formed based on offline real-world interpersonal connections, emotion can be as significant as technology in terms of affecting users’ political-participation intention, if not more important. 

## 3. Research Questions and Hypotheses

This study aims to answer the following question: Can we predict the willingness of Chinese citizens to participate in politics through the use of network groups? If so, what is the specific relationship between the two? Why does this association appear? Most of the literature on the impact of Internet use or social-media use on political participation in China supports the mobilization theory or the positive-effect theory [20,21,22,23,24]. The mobilization of social-network platforms is regarded as a realization mechanism of collective network action [33]. As an evolutionary form of online community in the era of social media, there are reasons why network groups continue with the mobilization logic in terms of public-participation willingness. Based on this, the first hypothesis (H1) is raised: Compared with those who do not join network groups, those who join network groups are more likely to generate a willingness to participate in politics. Political participation can be classified as direct participation and indirect participation according to whether there are intermediate links to help political participants influence the political process [43]. Since the China Social Survey 2019 questionnaire included both direct and indirect willingness to participate in politics, this study explores them separately. In this survey, direct willingness to participate in politics includes three parts, with the first part being participating in the major decision-making discussion of the village, neighborhood, or unit; the second part being participating in the election of the village (neighborhood) committee; and the third part being participating in online or offline collective-rights-protection actions. Indirect willingness to participate in politics covers another three sections, including the first section of reflecting social issues to media, the second section of reflecting opinions to government departments, and the third section of participating in voting for representatives of the people’s congresses at the district and county levels. Research has found that Chinese citizens’ willingness to participate directly in politics increased [26]. The second hypothesis (H2a), related to the willingness of direct political participation, is that compared with those who do not join network groups, those who do join network groups are more likely to generate a willingness to participate directly in politics. Another hypothesis of the study (H2b) is that compared with people who do not use network groups, people who use network groups will be more likely to have a willingness for indirect political participation. 

## 4. Research Methods

This study used data from the China Social Survey (CSS) 2019 conducted by the Institute of Sociology at the Chinese Academy of Social Sciences. The 2019 survey was the first time respondents were asked about their participation in online-network groups in the last two years. The survey did not specifically point out which social-media applications were counted. According to the researchers’ understanding, online-network groups mainly refer to popular social-media applications establishing online connections with others through a group form, such as WeChat, Weibo, QQ, and so on. According to the needs of the research questions, the researchers chose to use the data related to network groups and political-participation willingness. The researchers cleaned up the data with obvious nonstandard responses and a large number of missing values. After cleaning up the data, the number of samples (N) was 6629. The dependent variable of political-participation willingness in this study was classified into binary dependent variables. Each observation was independent of each other. The sample size was more than 15 times the number of independent variables. There was no multiple collinearity between independent variables (VIF < 5), obvious outliers, leverage points, or strong influence points. Based on the research questions and the characteristics of the data, this study selected the method of hierarchical logistic regression. As each hierarchical logistic-regression model used in the study passed the goodness-of-fit test (Hosmer–Lemeshow test was greater than 0.05), each regression model used in the study was effective. The specific control variables, independent variables, and dependent variables used in this study were as follows:

### 4.1. Control Variables

Several studies found that some demographic and socio-economic factors have an impact on citizens’ political participation, including education [44], income [45], living environment [46], socio-economic status [24], etc. Considering the available data and existing findings, the control variables included gender (M 0.46, SD 0.50), year of birth (M 1976.17, SD 14.12), education level (M 3.18, SD 2.17), political status (M 0.35, SD 0.68), and respondents’ judgment of their socio-economic status (M 1.39, SD 0.92). 

### 4.2. Independent Variable

The independent variable in this dimension was a binary variable, where 1 meant having joined specific network groups in the last two years, otherwise the value was 0. For example, the survey asked: Have you joined your relatives’ online-network groups in the last two years? If the answer was “yes,” it was coded as 1. If the answer was “no,” it was coded as 0. The China Social Survey 2019 only included the information of whether the participants joined specific online-network groups. It did not further investigate the extent to and frequency with which the participants used the online-network groups. The specific types of groups included relatives (M 0.66, SD 0.48), friends (M 0.64, SD 0.48), neighbors (M 0.26, SD 0.44), colleagues (M 0.42, SD 0.49), religions (M 0.02, SD 0.13), fellow villagers (M 0.20, SD 0.40), alumni (M 0.47, SD 0.50), interest groups (M 0.20, SD 0.40), public-welfare associations (M 0.10, SD 0.30), industries/associations (M 0.19, SD 0.39), rights protection (M 0.02, SD 0.13), and other groups (M 0.02, SD 0.13).

This study classified the network groups according to the main motivation of Chinese social interaction. Some studies believed that contemporary Chinese social interaction is mainly based on benefit–resource motivation [47]—that is, interest motivation is the main motivation of Chinese interpersonal interaction [47,48]. The research on the impact of network groups on people’s life satisfaction divides network groups into three categories: leisure–emotional relationship, business relationship, and public-welfare relationship [31]. To be more specific, this study further separated the leisure–emotional relationship into emotional relationship and leisure relationship so as to include four types of relationships: the emotional relationship, the leisure relationship, the business relationship, and the public-welfare relationship. This classification method not only has certain theoretical basis and foundation in the literature but also further clarifies the focus of each category to match the data. Among them, the emotional relationships include relative groups, friend groups, neighbor groups, classmate groups, alumni groups, and fellow-villager groups. The leisure relationship includes interest groups and religious groups. The business-relationship category includes colleague groups and industry or association groups. The public-welfare relationship includes public-welfare community groups and rights-protection groups.

### 4.3. Dependent Variable

The corresponding response in the survey was coded as a binary variable: 1 meant willing to participate and 0 meant unwilling to participate. Specific items related to the willingness to participate in politics included reflecting social issues to newspapers, radio, online forums, and other media (M 0.57, SD 0.50); reflecting opinions to government departments (M 0.62, SD 0.49); participating in the election of the village (neighborhood) committee (M 0.60, SD 0.49); participating in the major decision-making discussion of the village or unit (M 0.61, SD 0.49); participating in online or offline collective-rights-protection actions (M 0.58, SD 0.49); and participating in voting for the representatives of the people’s congresses at the district and county levels (M 0.70, SD 0.46).

## 5. Results

The following two tables present the results of hierarchical logistic regression of both direct and indirect political participation. 

### 5.1. Emotional Relationship

The results of the hierarchical logistic-regression models show that some network-group variables under specific categories were significantly correlated to political-participation intention (Table 1 and Table 2). The network groups that were significantly correlated to the generation of political-participation intention were mainly concentrated in the emotional relationship. Among them, the representatives included neighborhood groups and alumni groups. Whether to join neighborhood groups showed statistically significant differences in five of the six dimensions of political-participation willingness (*p* < 0.05). Specifically, keeping other variables unchanged, compared with those who had joined neighborhood groups, the probability of those who had joined neighborhood groups being willing to participate in the major discussions of their village or unit was 22.4% higher, and the probability of those who were willing to participate in the election of the village (neighborhood) committee was 32.1% higher in the category of direct political participation. In the category of indirect political participation, the probability of generating the intention to reflect social problems to the media was 14.6% higher, the probability of generating the intention to reflect opinions to the government department was 18.8% higher, and the probability of generating the intention to participate in the voting of deputies to the people’s congresses of the districts and counties was 27.9% higher than those who had not joined neighborhood groups, keeping other variables constant. Whether to join alumni groups showed statistically significant differences in four of the six dimensions of political-participation intention (*p* < 0.05). Keeping other variables unchanged, compared with those who had not joined alumni groups, those who had joined this type of group were 23.3% more likely to be willing to participate in the major discussion of their village or unit and 20.3% more likely to be willing to participate in the election of the village (neighborhood) committee in the category of direct political participation. In the category of indirect political participation, the probability of generating the intention to reflect opinions to government departments was 20.4% higher and the probability of having the intention to participate in voting for representatives of the people’s congresses at the district and county levels was 17.7% higher than those who had not joined alumni groups. Compared with other emotional groups, joining relative groups showed a unique significant negative correlation with the two dimensions of direct political-participation intention. Keeping other variables unchanged, the probability of those who had joined relative groups being willing to participate in the village (neighborhood)-committee election was 22.1% lower than those who had not joined this type of group, and the probability of generating willingness to participate in online or offline collective-rights-protection actions was 18.3% lower. Keeping other variables constant, people who had joined friend groups under the emotional-relationship category were 19.7% more likely to be willing to report their opinions to the government department, and the probability of generating intention to participate in online or offline collective-rights-protection actions was 21.9% higher than those who had not joined this type of group.

### 5.2. Leisure, Industrial, and Public-Welfare Relationships

In addition to emotional relationships, only a small number of subordinate groups of leisure relationships, industrial relationships, and public-welfare relationships showed certain predictive ability for some dimensions of political-participation willingness. The subordinate interest groups of the leisure relationship showed predictive ability in two of the six dimensions of political-participation willingness (Table 1 and Table 2). Keeping other variables unchanged, people who had joined interest groups were 20.9% more likely to report social problems to the media than those who had not joined interest groups, and the probability of willing to participate in online or offline collective-rights-protection actions was 22.9% higher. The sub-industry or association groups of the industry relationship showed certain predictive ability in one of the six dimensions of political-participation willingness. Keeping other variables unchanged, the probability of people who had joined sub-industry or association groups of the industry relationship category to be willing to participate in online or offline collective rights protection actions was 18.4% higher than those who had not joined these groups. Subordinate interest groups of public-welfare relationships showed predictive ability in two of the six dimensions of political-participation willingness. Keeping other variables unchanged, compared with those who had not joined the public-welfare-relationship groups, those who had joined the group were 69.7% more likely to be willing to participate in major decision-making discussions of their village, neighborhood, or unit and 75.8% more likely to be willing to participate in online or offline collective-rights-protection actions. Therefore, hypothesis H1 was verified in some network groups under certain categories: Compared with people who had not joined neighbors, friends, classmates and alumni, fellow villagers, interests, industries or associations, and rights-protection network groups, people who had joined these network groups were more likely to generate political-participation intention in some dimensions.

### 5.3. Political-Participation Willingness

From the perspective of each dimension of political-participation willingness, any one of the dimensions of political-participation willingness selected by this research had a significant correlation with at least two network-group variables (*p* < 0.05) (Table 1 and Table 2). The dimension of participating in online or offline rights-protection actions had a significant correlation with five network-group variables (*p* < 0.05), specifically including relative groups, friend groups, interest groups, industry or association groups, and rights-protection groups. There were four dimensions of political participation willingness that were significantly correlated with the three network-group variables (*p* < 0.05). Specifically, in the category of willingness to participate in direct political participation, the dimension of participating in major decision-making discussions of the village or unit was significantly positively correlated with the variables of neighborhood groups, alumni groups, and rights-protection groups (*p* < 0.05, B > 0). Also in the category of direct political-participation willingness, the dimension of willingness to participate in the election of the village (neighborhood) committee was significantly correlated with the variables of relative groups, neighbor groups, and alumni groups (*p* < 0.05). The willingness to participate in politics in this dimension had a significant negative correlation with the variable of relative groups (B < 0) but a significant positive correlation with the variables of neighborhood groups and alumni groups (B > 0). Therefore, hypothesis H2a was verified in some network groups under certain categories. The dimension of reflecting opinions to government departments in the category of indirect political participation was significantly positively correlated with the variables of groups of friends, neighborhood, and alumni (*p* < 0.05, B > 0). Also under the category of indirect political participation willingness, the willingness to participate in electing representatives of the people’s congresses at the district and county levels was significantly positively correlated with the variables of neighborhood groups, fellow-villager groups, and alumni groups (*p* < 0.05, B > 0). Therefore, hypothesis H2b was verified in some network groups under certain categories.

## 6. Analysis and Discussion

It is significant to understand the results comprehensively, as this empirical evidence indicates a potential new type of relationship between digital media, information, and people’s thinking. Previous studies typically approached the effect of digital media through people’s action without considering those who did not behave differently but started to think different after their access to digital media and the information contained there. Among others, political-participation intention is an important aspect of people’s thoughts about politics. The explanation between the significant correlation between the use of specific types of online-network groups and political-participation intention can not only fill this gap but also enrich our understanding of the political-mobilization function of digital media. The following discussions unpack the explanations from perspectives including virtual social connection, strong relationships, and social classification, community, and social connection.

### 6.1. Virtual Social Connection

Although we should take a cautious attitude towards technological determinism, without the role of Internet media and the content they communicate, there would be no significant difference in certain dimensions of the willingness to participate in politics when comparing joining certain network groups with not joining them. As the connections with relatives, alumni, and other groups have already been established offline in real life, the existence of these relationships is not affected by joining network groups in these categories. In a newly developed multi-dimensional taxonomy of political participation, “digitally networked participation” is a specific type of political participation, together with other types covering voting, institutionalized participation, protest, civic participation, and consumerist participation [6]. This type of participation features mutual connection established through digital communication technology. Having only strong relationships in real life but no virtual social connection built by network groups cannot form the sufficient condition for generating political-participation willingness. On the other hand, network groups create and maintain a virtual social connection, which requires a deep and strong real social connection as the premise to sufficiently mobilize political-participation willingness. Network groups that are significantly correlated with the intention of political participation all have a certain foundation of social connection in reality. It requires a combination of social connection in reality and virtual social connection to successfully trigger political-participation willingness. Network groups provide technical possibilities and practical conditions for the formation and maintenance of diversified online-network communities. As a specific form of network media, network groups create certain conditions for generating political-participation intention through the virtual social association established by information interactions inside them.

Although affirming the possible role of network groups and information dissemination inside them, the specific mode of the effect of online communication may vary according to the characteristics of the specific communication field constructed by the Internet. A classic study of online political communication found that the political effect of online communication is mainly about strengthening people’s existing political views rather than changing their views and decisions. Bruce Bimber and other researchers discovered the limited effects of government websites in the election process of the U.S., which mainly strengthen the views of voters who have already decided to support a certain candidate and cannot mobilize those who choose not to participate in the voting or persuade wavering people [12]. In terms of the use of social media in China, researchers have found that the offline and online political participation of social-media users in mainland China, Hong Kong, and Taiwan can be predicted by sharing political information and connecting with public actors [24]. Compared with government websites and social-media use, network groups provide a specialized two-way communication platform that citizens can choose to join and communicate with specific groups more conveniently and interactively. Even if not all types of network groups have strong effects, this study supports the limited effects of network groups on political-participation intention. With the virtual connection built by network groups, the possible limited effects have expanded from the strengthening of political conceptions and the generation of political-participation behavior to the new form of generating political-participation intention. Network groups more closely and conveniently link people with other people, and people with specific groups. Their political effects show adjustment and change compared with the political effects of traditional Internet use.

### 6.2. Strong Relationships

Apart from the function of the Internet, one of the key factors that we cannot ignore is social relationships. As a specific form of mass communication in the Internet era [49], the limitations of mass communication also apply to online communication to a certain extent. Joseph Klapper argued that mass communication itself is not necessarily and not enough to influence the audience; its effect is transmitted by other factors [50]. Among possible factors, the perspective of social relationships helps to explain the results. Classical studies in the early stage of mass communication have long been concerned with the impact of social relationships on people. Stanley Baran and Dennis Davis argued that people may be more influenced by others than by media [51]. Among many types of relationships, the closeness of the relationship will also play a key role in the occurrence of the impact. Paul Lazarsfeld’s research of the impact on voters in political elections in the American context found that voters’ voting behavior was the same as those closest to them but not as what the media told them [52]. The influence of this interpersonal relationship on people’s judgment and cognition has continued in the Internet era. Cass Sunstein believes that people believe something only because other people related to them also believe it [53]. Another perspective of understanding interpersonal relationships is the factor of emotion, which may help to explain the reason why strong relationships tend to correlate significantly with political-participation intention. In fact, the relationship between emotion and politics has already attracted some scholars’ interest. Recently published academic works discuss how emotion can be used as an effective tool for politicians’ purposes and goals, such as the citizen polarization created by emotional narratives in speeches of politicians [40], positive-emotion creation through fear and anger legitimization by the UKIP, and framing reactions to social worry and concern [41].The impact of social relations is significant in Western society, where the relationships between people are relatively simple, let alone in Chinese society, where interpersonal relationships play a significant role.

Chinese people are particularly vulnerable to the influence of strong relationships that have been established in real society, which also applies to the context of political-participation intention. Lan Peng argued that network groups reflect social relationships established in the real world, in which the strong relationship plays an important role [33]. The study of political mobilization based on the context of Western countries has also found that successful political mobilization depends on strong face-to-face interaction and communication between members [54,55]. This face-to-face interaction is, to a large extent, a characteristic of strong offline relationships. This leads to asking why strong relationships can play an important role. The formation of strong relationships is often accompanied by common social experience, which can bring a certain degree of social stickiness [53]. Cass Sunstein explained the cause of aggregation among members of strong relationships through the perspective of common social experience, and Ferdinand Tönnies further explained it from the perspective of community and how the consistent thinking mode of members is formed after aggregation. As a form of network community [33], network groups also have the characteristics of a community to a large extent. Under the concept system of community, Ferdinand Tönnies used the concept of “common understanding” to describe the unique and consistent beliefs of a community. The realization of “common understanding” needs to be based on the mutual understanding among the members of the community. The similarity of each other’s life experience makes each other’s way of thinking coordinated and similar [56]. Network groups facilitate the interaction and information exchange within the experiences of community members that have formed offline. On the basis of the common social experience that has been established in offline life, the interactions within network groups further deepen this “common understanding” and converge their willingness to participate in politics.

Another characteristic of strong relationships is the relatively closed and solidified structure. Although such characteristics weaken the vitality of the group, a relatively stable relationship is better than an open and mobile structure when influencing people’s judgments, opinions, and even actions [33]. Among the many types of social relations built by a network-circle group, social relationships linked by emotion can be relatively strong relationships. Whether it is blood relatives as the emotional link, friends as the emotional link, classmates with a common learning experience as the emotional link, or fellow villagers with a geographical relationship as the emotional link, they are all strong relationships with certain kinds of emotion as the links. The results of this study show that these strong relationships were significantly correlated to some dimensions of political-participation willingness. From the perspective of social capital, strong relationships in Chinese network groups have the characteristics of cohesive social capital to some extent. Robert Putnam regarded the relationship network as a kind of social capital. The members of the relationship network achieve mutual benefit through coordination and cooperation. He divided social capital into cohesive social capital and bridging social capital. Cohesive social capital is characterized by homogeneity, introversion, and strengthened internal identification, whereas bridging social capital features heterogeneity, extroversion, and transcendence of various social groups [14]. Internet users join emotional-relationship network groups according to the strong relationship network that has been established offline. Information is shared and disseminated within each network group. However, it is difficult to spread the information across network groups established by other individuals. The people who belong to a certain type of network group pay attention to internal identification and tend to think homogeneously. Therefore, the interaction of a network group of emotional relationships enhances the cohesive social capital, but it cannot substantially improve the bridging social capital. In a network group connected by various types of relationships, it is easier for people to build trust relationships in online virtual relationships based on the emotional relationships established offline. The trust between people in the virtual association of a network group may further bring about the acceptance of the spread of information in this type of network group. This, to some extent, explains why the network groups that can predict political-participation intention were mainly concentrated in the emotional-relationship category.

While highlighting the importance of strong relationships in the context of network groups, what makes the question complex is that different types of emotional-relationship network groups presented different forms of correlation with political-participation willingness. Although the majority of emotional groups that were significantly correlated with political-participation intention were positively correlated with political-participation intention, the relationship between relative network groups and political-participation intention was unique. Compared with those who had not joined a relative network group, those who had joined this type of group were less likely to generate willingness in the two dimensions of direct political participation. In other words, joining a network group of relatives with the strongest relationship among subordinates in the emotional-relationship category may inhibit the willingness to participate directly in two dimensions of politics. Relatives and families are important forms of personal social capital, and the blood relationship they represent is the initial manifestation of the community [56]. The unique correlation between the strongest emotional group and political-participation willingness deserves further study. In the context of network groups, the strength of the relationship and political-participation willingness does not present a monotonous linear relationship.

### 6.3. Social Classification, Community, and Social Connection

It is worth noting that some network groups that had not established strong relationships were also significantly related to the generation of political-participation willingness in some dimensions. Although there is evidence supporting that social-media exposure distracts participation, as it has a positive correlation with low-effect political participation [19], the use of some online-network groups was positively correlated with the emergence of political-participation intention—at least, evidence supports this argument in the context of Chinese online-network groups. Some groups under the categories of leisure relationships, industrial relationships, and public-welfare relationships showed certain predictive ability in some dimensions of political-participation willingness. Network groups represent specific categories of social groups, which may have certain impacts on individuals belonging to these groups. According to the social-classification theory, the behavior of specific groups and aggregates in the face of a given stimulus is more or less consistent [57]. The basic assumption of this theory is that despite the heterogeneity of modern society, people with many similar characteristics are highly likely to have similar ways of mass communication and thinking. These similar orientations and behavior patterns connect these people with mass media in a fairly uniform way. The concept of “community” can explain how individuals are affected by groups, which is similar to the meaning of social-classification theory but from a different perspective. The concept of “community” refers to the unified state of human will [56]. Under the environment of network groups, it is not only applicable to the network group in the context of strong relationships but can also explain to some extent the reason why network groups that have not established strong relationships are significantly correlated to political-participation willingness. In the process of explaining the concept of “community,” Ferdinand Tönnies put forward “the will field of community,” which argues that the more individuals are integrated with others in the same community, the less they can control their own will and ability [56]. Similarly, Zygmunt Bauman argued that the reason why the community is desirable is that it is extremely enticing to promise simplification: bringing maximum similarity and minimum diversity to individuals in the community [58]. The homogeneity formed after the simplification within the community brings comfort and stability to the individual and the collective and eliminates the fear, uneasiness, and anxiety caused by the uncertainty of life. The results of this study show that even if network groups bring new forms of network organization for specific social members, the social-classification theory and the concept of community can still continue to explain the characteristics, phenomena, and possible effects of network groups in the context of communication to a certain extent. The rights-protection network groups belonging to public-welfare relationships were significantly correlated to the willingness to participate in the major decision-making discussions of the village or unit where they are located and direct political participation in online or offline collective-rights-protection actions. Compared with those who had not joined the network groups of rights protection, those who had joined were 69.7% more likely to be willing to participate in the major decision-making and discussions of their village or unit. The probability of generating willingness to participate in online or offline collective-rights-protection actions was 75.8% higher. According to the social-classification theory, the homogenization mode of thinking exists in specific categories of social groups. People who choose to join network groups of rights protection generally have a strong sense of rights. Therefore, the possibility of generating political-participation intention is relatively high. Lan Peng defined “network community” as a relatively stable group of people connected by certain commonalities [33], and a network group is a specific form of network community. In the community, individuals feel warm, comfortable, and secure, and understand and trust each other without confusion [58]. As individuals in the network community have certain homogeneity, they are vulnerable to the influence of collective consciousness to a certain extent. A typical example is that the people who had joined the rights-protection groups in the above were more inclined to have some dimensions of political participation than those who had not joined. Similar to the social-classification theory, Cass Sunstein argued that there is an information-driven “social connection.” When individuals lack information, others’ statements and behaviors will become the basis for individual judgment [53]. The characteristic of this social connection is that information is transmitted through the connection of social groups, rather than professional information-communication institutions or scientific argumentation. This feature gives the information-driven “social linkage” certain limitations in the flow and effect of information: The authenticity and reliability of information cannot be agreed on and guaranteed for the whole society. As information only flows within social groups, such social linkage may cause one group to believe something, whereas another group believes the opposite [53]. Although the thoughts of social classification, community, and social connection help to explain how network groups, which are significantly related to the willingness to participate in politics, can play a possible role in political mobilization, these ideas also have their limitations. There are also several network groups that were not significantly related to the generation of political-participation willingness. In the context of network groups, the remaining results cannot be explained and supported by these concepts.

### 6.4. The Separation of the Social Category 

Network groups to some extent verify the social classification in the specific environment of the network, but at the same time strengthen the separation between different social categories. Individuals under the same social classification obtain a closer online virtual association through network circle groups, but it is difficult to establish a connection between individuals belonging to different social categories in the environment of network groups. Manuel Castells emphasized the networking logic brought on by the development of information technology for society by arguing that a network is a group of interconnected nodes, which are located at the intersection of the curve and the individual [1]. This networked way of thinking outlines a desirable, idealized, and interconnected world, but the specific field of network groups shows its unique characteristics of connection. Although network-group users connect within each group, there is no ideal interconnection curve between network groups in the real world. This to some extent violates the openness and connectivity of the network society described by Manuel Castells. The use of network groups has created an increasingly personalized media field. Compared with the earlier network community (*wangluo shequn*) in China, which is dominated by topics and builds links between members through content, network groups are mainly dominated by the interests, hobbies, and emotions of members [33]. Although satisfying individual community and information preferences, it also separates individuals from other types of groups and their related information. Despite feeling the warmth brought by network groups as communities, individuals in network communities may also pay a price at the thinking level. As a kind of network community, network groups continue the shortcomings of the traditional community: lack of motivation to reflect, criticize, and experiment [59]. The frequent and comprehensive exchanges among the people in the community means that they rarely consider information outside the community carefully. The development of information and communication technology is believed to contribute to the realization of wider information sharing and interconnection, which may influence and change the community’s acceptance of external information and the formation of the community members’ way of thinking. Zygmunt Bauman believes that the development of information and communication technology has disintegrated the common understanding formed naturally within a community [58]. This view is an overly optimistic imagination of a perfect interconnected world brought about by the development of communication technology, but it does not carry out targeted in-depth exploration and demonstration of the impact of a specific application of information-communication technology. There is the convenience of information communication within network groups and a hindrance of information flow between different network groups. A common understanding within the network community may still form and continue. The network groups formed according to the differentiation of social groups seem to have become an information-filtering mechanism, which is a customized and tailored information field for individuals. The voices heard and information read by individuals come from the social groups they choose. Individuals choose the communication system into which they want to integrate and join network groups to which they are close and in which they are interested. However, they cannot hear the voices of other network groups that are different from their own environment, interests, and hobbies. This explains to some extent why specific network groups, such as groups of colleagues and religious groups, could not predict the political-participation willingness in any dimension. In the context of deepening the separation of network groups, these network groups, formed for the purpose of disseminating professional information and sharing religious beliefs, may rarely involve public-affair discussions, and thus are unable to mobilize users’ willingness to participate in politics.

## 7. Conclusions

This study starts from the use of network groups, a specific online application, and provides new evidence and analysis from the perspective of political-participation intention for the discussion of the political effect of Internet use. It further extends the understanding of the political effect of Internet use in the context of contemporary China. Based on the analysis of the survey data, this study found that the network groups that were significantly correlated to the generation of political-participation intention were mainly concentrated in the emotional-relationship category, whereas only a few subordinate network groups with a leisure relationship, an industrial relationship, and a public-welfare relationship showed certain predictive ability in the individual dimensions of political-participation intention. Factors including Internet communication technology, social relations, and social groups jointly established and explained the relationship between the use of network groups and political-participation willingness. The information spread by a specific type of network group, the type of relationship among the members of the groups, and social groups may affect the generation of political-participation willingness. The discussions of the types of social relationships and social groups make the possible political effects of the network media in this paper not only limited to the network-communication technology itself but also extend to the people that the network-communication technology allows users to be associated with and the specific ways to connect with these people. At the level of network-communication technology, the virtual social connection established by the information interaction within network groups creates certain conditions for the generation of political-participation intention. The networking-connection mode realized by the network technology is further explored in terms of possible effects. It is no longer only concerned with strengthening people’s existing political concepts or generating political participation behavior but can also predict the generation of users’ political-participation willingness to a certain extent.

At the level of social relationships, the virtual social connection built by network groups allows the network-communication technology to form a possible synergy with the real social relationship. In the media field of network groups, the generation of the user’s political-participation intention is closely related to the strong relationship that has been established in real society. This is reflected in the data-analysis results, in which the network groups that could predict political-participation intention were mainly concentrated in the emotional-relationship category. Users of network groups are more likely to establish mutual trust in an online virtual association based on a strong offline relationship. This established online trust brings acceptance and recognition of the information spread in this type of network group and generates political-participation willingness on this premise. It is worth noting that among the network groups with the same type of strong relationship, different types of network groups were differently correlated to political-participation willingness. Although most of the emotional-relationship groups that were significantly correlated to political-participation intention displayed a positive correlation with political participation intention while keeping other variables unchanged, joining a relative group could predict the decline of the possibility of generating political-participation intention. The interpretation of this phenomenon requires future research focusing on this issue.

From the perspective of social classification, network groups represent specific types of network social groups, and these social groups may have certain impacts on individuals. People who join the same network groups belong to specific social groups and aggregations. Thus, the network communities formed by these people can often show similar ways of mass communication and thinking. This kind of homogenized thinking mode under the same social category helps to explain why some network groups of leisure relationships, industrial relationships, and public-welfare relationships could predict some dimensions of political-participation willingness. Although the explanatory power of the social-classification theory is recognized, this study also clearly acknowledges that network groups have strengthened the separation between different social categories while forming social classification. Different social categories cannot establish information sharing and effective links in the context of network communities, which often put information from outside at a disadvantage. Individuals join specific types of network groups according to their kinship, preferences, and interests. The information they receive is also personalized and shaped so that they cannot hear the voices of other social groups that are different from their environment and interests. This separation between social categories also explains to some extent why the network groups of colleagues and religions formed for the purpose of disseminating professional information and sharing religious beliefs could not predict political-participation willingness.

From the perspective of network communication, social relationships, and social classification, this study discusses the relationship between the use of network groups and the willingness to participate in politics in the context of contemporary China. Due to the limitation of the research scope and data, the independent variables focused on in this study only relate to whether individuals join specific network groups and do not include the degree of use of those groups and the characteristics of the information disseminated within them. If the relevant survey data in the future are further enriched to cover this information, we can discuss the political effects of using network groups from more dimensions. It would be interesting to investigate whether we can predict political-participation intention from the use extent of different categories of online-network groups. The effort to verify the potential causation relationship between the use of online-network groups and political-participation intention is also valuable. In addition, future endeavors can also be directed to explore which factor matters more in triggering Chinese citizens’ political-participation intention: the interpersonal connection established through the internet or the strong relationship. The unique and significant negative correlation between the relative network groups and the willingness to participate in politics in the kinship relationship also requires further explanation by future research with relevant evidence.

## Figures and Tables

**Table 1 behavsci-13-00302-t001:** Results of hierarchical logistic regression of direct political participation.

	Participating in the Major Decision-Making Discussion of the Village, Neighborhood, or Unit		Participating in the Election of the Village (Neighborhood) Committee		Participating in Online or Offline Collective-Rights-Protection Actions	
	B	OR	B	OR	B	OR
Control Variables						
Gender (male)	0.206 ***	1.229	0.024	1.024	0.086	1.09
Year of birth	−0.003	0.997	−0.006 *	0.994	0.006 *	1.006
Education Level						
Primary school (vs. no education)	0.246	1.279	0.447 **	1.563	0.138	1.148
Junior high school (vs. no education)	0.306 *	1.358	0.473 **	1.605	0.206	1.229
Senior high school (vs. no education)	0.136	1.146	0.351 *	1.42	0.159	1.172
Technical secondary school (vs. no education)	0.043	1.044	0.042	1.043	0.069	1.072
Vocational high school (vs. no education)	0.469	1.598	0.346	1.413	0.691 *	1.996
Junior college (vs. no education)	0.161	1.175	0.173	1.188	0.11	1.117
Undergraduate (vs. no education)	0.277	1.319	0.276	1.318	0.039	1.04
Postgraduate (vs. no education)	0.031	1.031	−0.087	0.917	−0.015	0.985
Political Status						
Chinese Communist Party member (vs. the masses)	0.193 *	1.213	0.197	1.218	−0.077	0.926
Communist Youth League member (vs. the masses)	0.261 *	1.298	0.340 **	1.405	0.249 *	1.283
Democratic parties (vs. the masses)	1.049 *	2.854	0.577	1.78	0.01	1.01
Socio-Economic Status						
Upper layer (vs. lower layer)	0.019	1.02	0.064	1.067	0.094	1.098
Upper-middle layer (vs. lower layer)	0.072	1.075	0.146	1.157	0.028	1.029
Middle layer (vs. lower layer)	0.053	1.054	0.236	1.266	0.013	1.014
Lower-middle layer (vs. lower layer)	0.604	1.83	0.633	1.883	0.640 *	1.897
Network-Group Variables						
Emotional Relationship						
Relatives (vs. no)	−0.073	0.93	−0.250 **	0.779	−0.202 *	0.817
Friends (vs. no)	0.12	1.127	0.161	1.175	0.198 *	1.219
Neighbors (vs. no)	0.202 **	1.224	0.278 ***	1.321	0.092	1.097
Alumni (vs. no)	0.210 **	1.233	0.185 *	1.203	0.084	1.088
Fellow villagers (vs. no)	0.109	1.115	0.161	1.174	0.123	1.131
Leisure Relationship						
Interest groups (vs. no)	0.075	1.078	−0.107	0.898	0.206 **	1.229
Religions (vs. no)	−0.096	0.909	−0.244	0.784	0.038	1.038
Industrial Relationship						
Colleagues (vs. no)	0.002	1.002	0.035	1.036	−0.047	0.954
Industries/associations (vs. no)	0.096	1.101	0.078	1.082	0.169 *	1.184
Public-Welfare Relationship						
Public-welfare associations (vs. no)	0.163	1.177	0.219	1.245	0.011	1.011
Rights protection (vs. no)	0.529 *	1.697	0.1	1.106	0.564 *	1.758
Other groups (vs. no)	−0.077	0.926	−0.093	0.911	−0.005	0.995
Constant	6.384	592.474	12.56	284,844.77	−12.003	0

Note: * *p* < 0.05,** *p* < 0.01,*** *p* < 0.001.

**Table 2 behavsci-13-00302-t002:** Results of hierarchical logistic regression of indirect political participation.

	Reflecting Social Issues to Media		Reflecting Opinions to Government Departments		Participating in Voting for the Representatives of the People’s Congresses at the District and County Levels	
	B	OR	B	OR	B	OR
Control Variables						
Gender (male)	0.177 **	1.193	0.224 ***	1.251	0.206 **	1.229
Year of birth	0.006 *	1.006	0.005 *	1.006	−0.008 **	0.992
Education Level						
Primary school (vs. no education)	0.061	1.063	0.129	1.138	0.058	1.06
Junior high school (vs. no education)	0.111	1.118	0.165	1.179	−0.017	0.983
Senior high school (vs. no education)	0.135	1.145	0.135	1.145	0.09	1.094
Technical secondary school (vs. no education)	−0.077	0.926	−0.117	0.889	−0.064	0.938
Vocational high school (vs. no education)	0.404	1.497	0.317	1.373	0.186	1.205
Junior college (vs. no education)	0.142	1.153	0.123	1.131	0.068	1.07
Undergraduate (vs. no education)	0.232	1.261	0.231	1.26	0.162	1.176
Postgraduate (vs. no education)	0.383	1.466	0.248	1.281	0.492	1.635
Political Status						
Chinese Communist Party member (vs. the masses)	−0.032	0.969	0.152	1.164	0.274 *	1.315
Communist Youth League member (vs. the masses)	0.275 *	1.316	0.420 ***	1.522	0.475 ***	1.608
Democratic parties (vs. the masses)	0.769	2.158	0.974	2.648	1.213	3.362
Socio-Economic Status						
Upper layer (vs. lower layer)	0.051	1.052	0.042	1.043	−0.017	0.983
Upper-middle layer (vs. lower layer)	0.028	1.028	−0.006	0.994	0.009	1.009
Middle layer (vs. lower layer)	−0.125	0.883	−0.017	0.983	0.075	1.078
Lower-middle layer (vs. lower layer)	0.687 *	1.987	0.662 *	1.939	0.614	1.847
Network-Group Variables						
Emotional Relationship						
Relatives (vs. no)	−0.122	0.885	−0.106	0.899	−0.087	0.917
Friends (vs. no)	0.089	1.093	0.180 *	1.197	0.053	1.055
Neighbors (vs. no)	0.136 *	1.146	0.172 *	1.188	0.246 **	1.279
Alumni (vs. no)	0.126	1.135	0.186 **	1.204	0.163 *	1.177
Fellow villagers (vs. no)	0.12	1.128	0.09	1.094	0.214 *	1.239
Leisure Relationship						
Interest groups (vs. no)	0.190 *	1.209	0.08	1.083	0.095	1.1
Religions (vs. no)	−0.065	0.937	0.024	1.025	−0.246	0.782
Industrial Relationship						
Colleagues (vs. no)	−0.072	0.93	−0.022	0.978	−0.029	0.972
Industries/associations (vs. no)	0.107	1.113	0.113	1.119	−0.011	0.989
Public-Welfare Relationship						
Public-welfare associations (vs. no)	0.186	1.204	0.131	1.14	0.029	1.03
Rights protection (vs. no)	0.417	1.518	0.361	1.434	0.252	1.287
Other groups (vs. no)	−0.06	0.942	−0.129	0.879	−0.218	0.804
Constant	−12.197	0	−10.888	0	16.16	10,429,144.9

Note: * *p* < 0.05,** *p* < 0.01,*** *p* < 0.001.

## Data Availability

Publicly available datasets were analyzed in this study. This data can be found here: http://css.cssn.cn/css_sy/.

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
