# Peer review of "Online-Network-Group Use and Political-Participation Intention in China: The Analysis Based on CSS 2019 Survey Data"

_behavsci, 2023, doi:10.3390/bs13040302_

Round 1

Reviewer 1 Report

The topic is interesting, the aims of the article are clearly presented, methodology is adequate

The article should be improved from two main points of view:

1. the concept of "online network group" is not clear: both in the presentation of regression and in the theoretical part, the author should clarify what these groups are, and what is their connection with offline correspondent groups - if relevant; the regression tables in particular at present are quite unclear at present

2. there is a huge amount of literature about the connection between social networks, even the digital ones, and attitudes towards politics and participation; given that the article focusses only on the Chinese context, a dialogue between its empirical data and other research focussed on the same processes in different territorial contexts should be developed in the article, in order to distinguish which of the results presented by the authors are distinctive of the Chinese context and which are more general

Reviewer 2 Report

This is an interesting paper, mainly for its contribution to knowledge on moder china and its social networking. The paper is written in excellent language, it has a sound literature review and the research method is very good also. The discussion section is also thorouglhy prepared and presented.

It is not clear if the authors participated in the China Social Survey (CSS) 2019 or just use its results. It would be interesting to know which social media applications were counted in this research and to what extend. Furthermore, what is the volume of users per social network application would be important to know. 

Furthermore, the 'network' or 'social network' terms raise some questions, since this is a 'standard' statistical analysis, and does not include any complex network science ideas. For example, 'network groups' or 'Strong relationships' (vs 'weak relationships' or is it 'weak ties') are present in the Social Network Analysis Literature for many years, alas, not found in the present paper. The same applies for the term 'community' which is quite an important cohesive group in Social Networks (see Newmann), but is not presented in such a way here. I would suggest that the aurhors should be more careful with such terms. 

In a final observation, I would like to point out that (to a 'western' reviewer), some ways of living and socially acting in China are really obscure. Indeed, in some cases there may be a bias in our knowledge, which may come from propaganda of some sort. For example, what is meant by 'political participation willingness' ? To what kind of elections? How many different (and opposed) political parties are present in these elections? Is such kind of participation kind of obligatory, if one wants to proceed in his/hers profession or social status? Some discussion should be made here.

Reviewer 3 Report

Thanks for giving me the chance to read this very interesting article.

The manuscript focuses on the relationship between the use of online network group  and the political participation intention in contemporary China.

The topic is very relevant, the methodology is appropriate and results, showing that although most of the online network groups are positively correlated with political participation intention, the possibility of generating political participation intention of those who join the relative network group is significantly lower than those who do not, are interesting.

Thus this work can be considered a relevant contribution to the broader literature looking into political participation and social network.

However the study presents some flaws that should be fixed before its publication.

1) Literature review and theoretical background

First, the definition of “political participation is unclear. You use Huntington definition of 1976, but much have been written since then. I suggest to provide a clear definition (analyzing recent sources) since your definition of political participation is the basis of the article. Same when you define “intention” you should analyze relevant literature.

Second, you base your reflections about internet and political participation on classic works.

See “His subsequent research finds that Americans who read online news are less likely to par-55 ticipate in politics than others (Putnam, 2000). Delli-Carpini also has a similar view that 56 the increase of the Internet use time causes the reduction of the time of political participa-57 tion (Delli-Carpini, 2000). Some other scholars find that the use of the Internet will lead to 58 the dispersion of users' energy, which leaves the public little time to participate in politics 59 (Kaufhold, Valenzuela, & de Zúñiga, 2010)”.

Of course Putnam, etc are classics, but many recent studies have analyzed these topic and should be acknowledged.

See for instance:

Campante, F., Durante, R., & Sobbrio, F. (2018). Politics 2.0: The multifaceted effect of broadband internet on political participation. Journal of the European Economic Association, 16(4), 1094-1136.

But there is a wide literature on the topic.

Third and most important, you seem to acknowledge that emotion play a crucial role (you mention  “emotional relationship”). This is not only true, but a crucial fact in internet political discussion.

Again, it is seminal to discuss the role of emotion in internet political discussion in the theoretical background.

This work analyzing Brexit might help:

Cervi, L., & Carrillo-Andrade, A. (2019). Post-truth and Disinformation: Using discourse analysis to understand the creation of emotional and rival narratives in Brexit. ComHumanitas: Revista Científica De Comunicación, 10(2), 125-149. https://doi.org/10.31207/rch.v10i2.207

2) Conclusions

The discussion is good, but again, it is based on very “old literature. It should therefore be expanded comparing your results to the results of other recent works in different contexts .

GOOD LUCK!

The research finds that the specific online network groups that can predict the 13 political participation intention are mainly concentrated in the category of emotional relationship. Among them, The virtual connection 17 built by online communication technology, the social relations and the influence of social groups on 18 individuals can help to explain the correlation between them. 19

 this study uses the hierarchical logistic re-12 gression method.

Reviewer 4 Report

Thank you for the opportunity of reading and reviewing your interesting manuscript. It addresses a topic very interesting and relevant for the current ride of online use in all aspect of social life. The paper is well structured and well written, it includes a long Introduction section which covers the literature and theoretical background, too. Therefore I recommend either to separate a Literature section or to rename this first section as Introducction and theoretical background or similar.

Regarding the research part, the authors clearly present their aim and formulate research hypotheses. They use data from 2019 China Social Survey and test the hypotheses. The results of the econometric treatment are presented and then discussed in a separate section. I really appreciate this part and how the authors manage to discuss and retrieve valuable conclusions.

My final suggestion id to include explicit implications and future directions for investigations that were retrieved from the study.

And one formal remark, the authors need to follow the formatting and referencing style of the journal.

Good luck!

Round 2

Reviewer 3 Report

The paper has sufficiently improved and is now ready to be published.